# Prospective Association between Adherence to the Mediterranean Diet and Health-Related Quality of Life in Spanish Children

**DOI:** 10.3390/nu14245304

**Published:** 2022-12-14

**Authors:** Charlotte Juton, Paula Berruezo, Luis Rajmil, Carles Lerin, Montserrat Fíto, Clara Homs, Genís Según, Santiago F. Gómez, Helmut Schröder

**Affiliations:** 1Endocrinology Department, Institut de Recerca Sant Joan de Déu, 08950 Barcelona, Spain; 2Gasol Foundation Europe, 08330 Sant Boi de Llobregat, Spain; 3Pediatric and Public Health, 08023 Barcelona, Spain; 4Centro de Investigación Biomédica en Red de Diabetes y Enfermedades Metabólicas Asociadas (CIBERDEM), Instituto de Salud Carlos III, 28029 Madrid, Spain; 5Cardiovascular Risk and Nutrition Research Group, IMIM Hospital del Mar Medical Research Institut, 08003 Barcelona, Spain; 6CIBER Physiopathology of Obesity and Nutrition (CIBERobn), Instituto de Salud Carlos III, 28029 Madrid, Spain; 7Global Research on Wellbeing (GRoW), Blanquerna Faculty of Health Sciences, Ramon Llull University, 08022 Barcelona, Spain; 8CIBER of Epidemiology and Public Health (CIBERESP), Institute of Health Carlos III, 28029 Madrid, Spain; 9GREpS, Health Education Research Group, Nursing and Physiotherapy Department, University of Lleida, 25008 Lleida, Spain

**Keywords:** children, diet quality, health-related quality of life, prospective associations

## Abstract

Health-Related Quality of Life (HRQoL) is gaining attention in children and adolescents because it is an important outcome of their health status and well-being. Therefore, it is important to identify determinants for HRQoL. Currently, there is scarce and mainly cross-sectional evidence on the relationship between adherence to the Mediterranean diet and HRQoL in children and adolescents. Therefore, the objective of the present study was to assess the prospective association between adherence to the Mediterranean diet and HRQoL in Spanish children. The study was carried out in 1371 children aged 8 to 10 from different Catalan elementary schools with a medium follow-up of 15 months. The KidMed and KIDSCREEN-10 questionnaires were used to assess the relationship between diet and HRQoL, respectively. The KidMed score at baseline was positively associated with HRQoL (β = 0.320; 95% CI 0.101–0.540) after adjusting for confounders. Additionally, the logistic regression analysis showed positive associations between baseline consumption of fruit, vegetables, pulses, and high adherence to the Mediterranean diet and HRQoL at follow-up (*p* < 0.05 for all) while the consumption of fast-food, pasta or rice, baked good or pastries, and sweets were negatively correlated (*p* < 0.05 for all) with HRQoL at follow-up. In conclusion, adherence of the Mediterranean diet was predictive for HRQoL in Spanish children, but further prospective studies are needed to confirm this result.

## 1. Introduction

In 1948, health was defined by the World Health Organization as “a state of complete physical, mental and social well-being and not merely the absence of disease or infirmity” [1]. From this definition began to emerge the concept of health-related quality of life (HRQoL) which is used to assess the perception of an individual’s physical and mental health [2]. Widely applied to assess the quality of life in patients with chronic illnesses, it is also now an instrument of interest in the general population. In children and adolescents, perceived general well-being, including physical and psychological well-being, social interaction, and school performance, is particularly important for enjoying a fulfilling life and growing into adulthood confidently [3]. Unlike the Western diet, the Mediterranean diet (MD) diet is rich in phytochemicals, unsaturated fat, fibers, and antioxidants known to improve cardiometabolic health and reduce chronic inflammation, but a recent body of evidence has also shown its benefits for cognitive health and against the risk of Alzheimer’s disease, dementia, and depression [4,5,6,7]. Likewise, cross-sectional evidence indicated that greater adherence to a MD was correlated with higher HRQoL however, with regard to longitudinal studies, only one evaluated this relationship and found no significant association after controlling for confounding factors and baseline HRQoL [8]. While adolescence is a time of significant physiological and psychological changes, promoting a MD from childhood may improve their HRQoL and help them navigate this transition more easily. However, the current limited evidence is based mainly on cross-sectional evidence with its inherent limitations [9,10,11,12,13,14,15]. There is a need for more robust evidence which can be provided by prospective study designs [16]. Therefore, the objective of the present study was to assess the prospective association between adherence to the MD and HRQoL in Spanish children.

## 2. Materials and Methods

### 2.1. Study Population

This study was designed as a prospective study as part of a community model of childhood obesity prevention called POIBC. The full protocol of the POIBC study has already been published elsewhere [17]. Briefly, the POIBC study was a two-year parallel intervention study (2012–2014) to assess the effectiveness of the THAO-child health program. In this study, 2249 children aged 8 to 10 from different Catalan elementary schools (4th and 5th grade) were followed for approximately 15 months. After listwise deletion of the participants with missing values, 1371 children (681 girls and 690 boys) with a mean age of 10.1 ± 0.6 years old remained in the final sample size.

### 2.2. Health-Related Quality of Life Assessment

HRQoL was determined by the KIDSCREEN-10 questionnaire [18], which comprises 10 items that focus on the subjective perception of health and well-being. The children answered each question on a 5-point Likert scale summing up to a total global score ranging from 25.4 to 83.8. A Rasch analysis was performed according to the KIDSCREEN-10 manual [19]. This analysis identifies items that represent a one-dimensional global latent HRQoL trait. The KIDSCREEN-10 Item statements are: (1) Have you felt fit and well? (2) Have you felt full of energy? (3) Have you felt sad? (4) Have you felt lonely? (5) Have you had enough time for yourself? (6) Have you been able to do the things that you want to do in your free time? (7) Have your parent(s) treated you fairly? (8) Have you had fun with your friends? (9) Have you got on well at school? (10) Have you been able to pay attention? The cut-off of high HRQoL was equal or more than 53.1 units of the KIDSCREEN-10 score according to Ellert and colleagues [20].

### 2.3. Dietary Assessment

The KidMed index assessed children’s adherence to the MD [21]. It was elaborated based on the principles of the MD and got previously validated in a sample of Spanish children participating in the EnKid study [21]. The questionnaire is composed of 16 items, 4 indicating a weaker adherence equivalent to −1 and 12 items indicating a higher adhesion equivalent to +1, giving a total score ranging from −4 to 12. A higher score denotes better adherence to the MD. From a categorical point of view, a poor adherence represents a score lower than 4 points, a medium adherence a score between 4 and 8 points, and a high adherence a score higher than 8 points.

### 2.4. Physical Activity

Children’s physical activity levels were assessed by the Physical Activity Questionnaire for Children (PAQ-C) [22]. It includes 9 items, each of which can range from 1 to 5 points and the final score being the average of all the items. Greater scores denote higher levels of PA.

### 2.5. Maternal Socioeconomic Status

Maternal education level was collected as an indicator of socioeconomic status and categorized into 2 levels: (i) primary and secondary school, and (ii) university degree.

### 2.6. Anthropometric Variables

Anthropometric measurements were assessed for each child following a standard protocol by trained personnel. Body weight and height were measured by an electronic scale (SECA 813) and a portable stadiometer (SECA 213), respectively. BMI z-score was computed using age and sex-specific reference values from the World Health Organization (WHO) [23].

### 2.7. Statistical Analysis

Descriptive analysis of baseline variables according to HRQoL tertile distribution was carried out by general linear models. Polynomial contrast was used to estimate p for linear trend with a post hoc Bonferroni correction for multiple comparisons.

Prospective associations of baseline adherence to the MD with HRQoL at follow-up were performed with linear regressions. The final model was adjusted for sex, age, allocation to intervention group, school, maternal education, zBMI, and HRQoL at baseline. The assumption of normality in the regression models was assessed by normal probability plot.

Cubic spline analysis was conducted to determine nonlinear associations between baseline adherence to the MD and HRQoL at follow-up with the use of the “gam” package in R version 3.0.2.

The predictive value of each item and categories of the KidMed index with HRQoL was analyzed by logistic regression adjusted for sex, age, allocation to intervention group, school, maternal education, zBMI, and HRQoL at baseline.

Interactions of sex and age with HRQoL were tested. The associations were considered significant if *p* < 0.05. All statistical analysis was performed using SPSS for Windows version 22 (SPSS, Inc., Chicago, IL, USA).

## 3. Results

No significant interactions were observed for sex and KIDSCREEN-10 score and age and KIDSCREEN-10 (*p* > 0.05 for both).

Descriptive analysis revealed a positive association of baseline adherence to the MD and physical activity with baseline HRQoL (*p* < 0.001 for both) (Table 1).

Logistic regression analysis (Table 2) showed that baseline “takes a fruit or fruit juice every day” (*p* = 0.038), “has fresh or cooked vegetables regularly once a day” (*p* = 0.037), “likes pulses and eats them more than once a week” (*p* = 0.039), and high adherence to the MD (*p* = 0.001) were positively associated with HRQoL at follow-up after adjusting for sex, age, allocation to intervention group, school, maternal education, physical activity, zBMI, and HRQoL at baseline. Conversely, “eats > 1 meal/week in fast food restaurants” (*p* = 0.001), “consumes pasta or rice at least 5 days per week” (*p* = 0.035), “has commercially baked goods or pastries for breakfast” (*p* < 0.001), and “takes sweets and candy several times every day” (*p* = 0.012) were negatively correlated with HRQoL after controlling for the same confounders.

Prospective analysis (Table 3) showed that KidMed score was associated with HRQoL for all models (*p* < 0.001), adjusted for gender and age (model 1), then additionally for allocation to intervention group, school, maternal education, physical activity, zBMI (model 2), and finally also for HRQoL at baseline (model 3).

Dose response analysis (Figure 1), adjusted for sex, age, intervention group, school, maternal education, physical activity, zBMI, and baseline HRQoL revealed a significant linear (*p* < 0.001) but no significant non-linear association (*p* = 0.183) between adherence to the MD and HRQoL. The curve slightly flattened beyond 4 units of the KidMed index.

## 4. Discussion

In this study, we found that the KidMed score was prospectively associated with HRQoL after adjusting for gender, age, allocation to intervention group, school, maternal education, physical activity, zBMI, and HRQoL at baseline. Through the literature, eight studies assessed the relationship between the KidMed index and the KIDSCREEN score from which one study had a longitudinal design while the others were cross-sectional [9,10,11,12,13,14,15,16]. Esteban-Gonzalo and colleagues reported in a longitudinal study design, a significant and positive association between adherence to the MD and HRQoL in adolescent girls but the statistical significance disappeared after further adjustment for HRQoL at baseline [16]. The authors found also a non-significant positive relationship between adherence to the MD and HRQoL [16]. The absence of statistical significance might be explained by the fact that they had a considerably smaller sample size in their strata compared to the present study, which in turn reduced statistical power. Additionally, Esteban-Gonzalo and colleagues provided results on the subjective well-being based on another validated questionnaire, but these results were hardly comparable to ours because this questionnaire measured different aspects. Mixed results were also reported from cross-sectional studies [9,10,11,12,13,14,15]; five studies reported a positive significant association, one a positive non-significant association and one a negative non-significant association. Of the studies that found a significant positive association, all were concentrated in the Mediterranean region, including Greece, Spain, and Portugal [9,10,11,13,14]. Compared to our study, they were conducted in slightly older children, aged 11 to 18, and had a smaller sample size, with the largest reaching 956 participants [9,10,11,13,14]. Despite significant associations, most studies had little or no adjustment for confounders, with the exception of Evaristo and colleagues who adjusted for age, sex, pubertal stage, socioeconomic status, sleep duration, and BMI [10]. A non-significant positive association was found in a Chilean study but compared to our results, children had a lower baseline KidMed score possibly suggesting a change in the Chilean diet [12]. Finally, Mitri and colleagues found a non-significant negative association however, in the multiple regression analysis, the KIDSCREEN-27 score had a variance inflation factor >4 indicating a multicollinearity issue in the analysis [15]. Another prospective study of Australian adolescents aged 11–18 found a significant positive association between the MD diet and the emotional subscale of the PedsQL, but the questionnaire used to assess diet quality had not been previously validated [24].

Furthermore, we found that baseline consumption of fruit, vegetables, and pulses was also positively associated with HRQoL at follow-up while consumption of fast-food, candies, pastries, and pasta or rice reported the inverse relationship. To our knowledge, no prospective study has assessed these eating behaviors with HRQoL, nevertheless a few cross-sectional and cohort studies using other HRQoL-related questionnaires have provided insight [25,26,27,28,29]. With regard to fruit and vegetables intakes, a British study found that adolescents that consumed the optimal intake of fruit and vegetables (5 portions per day) had better score in physical functioning, emotional functioning, social functioning, psychosocial health summary score, and total score of the PedsQL after adjusting for covariates [25]. This association remained true during summertime for emotional functioning and social functioning although in both time periods the magnitude was small [25]. Another study found that fruit and vegetables servings per day were positively associated with the social functioning subscale of the PedsQL in a group of American adolescents, but the study had a relatively small sample size [26]. Regarding unhealthy foods, fast-food, candies, and pastries were negatively associated with a higher HRQoL. Likewise, a study reported a reduction in HRQoL utilities score in Australian adolescents that ate fruits and vegetables often compared to sometimes or rarely, or fast food or takeaway after adjustment [30]. We did not find any association between skipping breakfast and HRQoL, but two Japanese cohort studies indicated a negative association [27,28]. These studies used the COOP chart questionnaire and the authors stated that it had been previously validated in children, but no abstract was found in either the Japanese pediatric population or other pediatric populations [27,28]. With respect to dairy consumption, a five year cohort study found that Australian adolescent boys with a high consumption of yoghurt had a higher PedsQL’s psychosocial health summary and school functioning scores at follow-up, but this study presented a high attrition rate (>50%) [29].

Because the MD is rich in polyunsaturated and monounsaturated fat, phytosterols and water-soluble fibers it reduces cholesterol level and protect against cardiovascular diseases [5]. The MD diet also decreases oxidative stress, inflammatory markers, DNA damage and cell proliferation making it a powerful ally against cancer [31]. Regarding mental health, a few potential molecular mechanisms have been proposed. The first and second mechanisms that involve inflammatory state and redox balance are closely related and both have been described in patients with depression [32,33,34]. Pro-inflammatory diets are characterized by high intakes of refined carbohydrates and saturated fats and low intakes of antioxidants and poly and monounsaturated fats which may alter the redox balance [35]. This alteration can lead to a state of chronic cellular inflammation that affects neurotransmitters, neuroendocrine function and brain activity, and triggers mental health disorders [36,37]. The MD, characterized by reduced saturated fatty acid and amino acid intakes and increased phytochemical and fiber intakes, has been shown to reduce oxidative stress and inflammatory markers which can have a protective effect on the brain [5,38,39]. The third mechanism implicates the gut microbiota. Animal models fed on a high fat diet or high calorie diet showed a shift in gut microbiota population associated with anxiety, poorer cognitive flexibility, and impaired social and object recognition indicating a potential relationship between diet, gut microbiota, and brain [40,41]. The MD increases gut microbiome diversity, gut homeostasis, and microbiota-derived metabolites. In the elderly, microbiota enrichment following a MD diet was positively associated with improved cognitive function and reduced inflammatory markers [5,42]. A fourth mechanism is a dysregulation of the hypothalamic-pituitary-adrenal axis (HPA) which increases the level of cortisol, a frequent marker of patients with depression [43]. Intervention studies with vitamin C supplementation, omega 3 fatty acids or polyphenol rich food have been found to reduce cortisol level [44,45,46,47]. Furthermore, in a population of primates, a MD diet reduced the cortisol response to stress compared to primates fed on a Western diet [48]. A fifth potential mechanism comes from the brain-derived neurotrophic factor (BDNF) which also seems to be lowered in depressive patients [49]. In the PREDIMED study, depressive patients randomized to a MD supplemented with nuts reported lower risks of low plasma BDNF concentrations compared to the control group [50]. A final mechanism involving the tryptophan-kynurenine metabolism has been described due to its involvement in the production of serotonin, but its regulation as well as the availability of tryptophan are not yet fully understood [51].

The exposure and outcome in the present study were recorded by questionnaires and therefore subject to the inherent limitations of self-reported data, such as memory bias, misunderstanding, and social desirability. The strengths of the study were the prospective design and a relatively large sample size. Furthermore, a Rasch analysis was performed to calculate the overall one-dimensional HRQoL score. This procedure gives a more accurate estimate of the HRQoL in comparison with a simple Likert scale calculation that has been applied in several of the referenced papers.

## 5. Conclusions

In conclusion adherence to the Mediterranean diet was predictive for HRQoL in Spanish children. The consumption of fruits, vegetables, and legumes, characteristic of the Mediterranean diet, was positively associated with high HRQoL, while the consumption of fast food and pasta or rice, more characteristic of the Western diet, was negatively associated with high HRQoL. Further prospective studies are needed to confirm this result.

## Figures and Tables

**Figure 1 nutrients-14-05304-f001:**
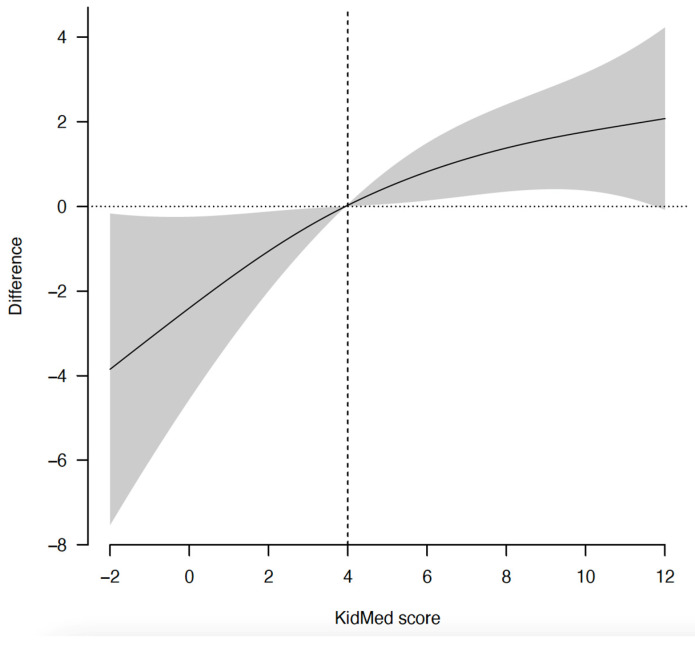
Dose response analysis between adherence to the Mediterranean diet and HRQoL. Reference value of the KidMed score was set to 4 units. Gray shadow = 95% confidence interval.

**Table 1 nutrients-14-05304-t001:** Baseline characteristics of the study population according to health-related quality of life ^1^.

	Total	1st Tertile	2nd Tertile	3rd Tertile	*p* for Linear
	*n =* 1371	*n =* 471	*n =* 492	*n =* 408	Trend
KIDSCREEN-10 ^2^	50.4 (25.4; 83.8)	41.4 (25.4; 45.7)	49.9 (46.9; 53.1)	61.6 (55.1; 83.8)	
Girls	681 (49.7)	237 (50.3)	240 (48.8)	204 (50.0)	0.908
Age (years)	10.1 (10.0; 10.2)	10.1 (10.0; 10.2)	10.1 (10.0; 10.1)	10.1 (10.0; 10.2)	0.506
Maternal education ^3^	468 (34.1)	143 (30.4)	171 (34.8)	154 (37.7)	0.021
zBMI	0.70 (0.64; 0.76)	0.76 (0.65; 0.86)	0.66 (0.55; 0.76)	0.68 (0.57; 0.79)	0.341
KidMed score ^4^	6.9 (6.7; 7.0)	6.1 (5.9; 6.3)	7.1 (6.8; 7.3)	7.5 (7.3; 7.7)	<0.001
Physical activity	2.98 (2.94; 3.01)	2.68 (2.22; 2.74)	3.04 (2.97; 3.10)	3.25 (3.18; 3.31)	<0.001

^1^ Values are presented as number (proportion) and mean (95% confidence interval or range in the case of KIDSCREEN-10, for categorical and continuous variables, respectively). ^2^ KIDSCREEN-10 ranged from 25.4 to 83.8 with higher score indicating higher HRQoL. ^3^ University degree. ^4^ KidMed score ranges from −4 to +8 with higher score indicating a higher adherence to the MD.

**Table 2 nutrients-14-05304-t002:** Prospective association of baseline categories of the KIDMED index and each of its item with high quality of life at follow-up ^1,2^.

	OR	95% CI	*p* Value
Takes a fruit or fruit juice every day	**1.31**	1.02; 1.69	0.038
Has a second fruit every day	1.19	0.94; 1.50	0.150
Has fresh or cooked vegetables regularly once a day	**1.29**	1.02; 1.63	0.037
Has fresh or cooked vegetables more than once a day	1.03	0.79; 1.34	0.849
Consumes fish regularly (at least 2–3 times/week)	1.21	0.95; 1.55	0.127
Eats > 1 meal/week in fast food restaurants	**0.59**	0.43; 0.81	0.001
Likes pulses and eats them more than once a week	**1.29**	1.01; 1.65	0.039
Consumes pasta or rice at least 5 days per week	**0.78**	0.62; 0.98	0.035
Has cereals or grains (bread, etc.) for breakfast	1.12	0.87; 1.45	0.365
Consumes nuts regularly (at least 2–3 times/week)	1.15	0.92; 1.45	0.230
Uses olive oil at home	1.04	0.72; 1.50	0.850
Skips breakfast	0.85	0.47; 1.52	0.572
Has a dairy product for breakfast (yoghurt, milk, etc.)	1.36	0.98; 1.89	0.068
Has commercially baked goods or pastries for breakfast	**0.56**	0.42; 0.79	<0.001
Takes 2 cups of yoghurts and/or some cheese (40 g) daily	1.19	0.91; 1.55	0.208
Takes sweets and candy several times every day	**0.65**	0.46; 0.91	0.012
Adherence to the Mediterranean diet			
Low	1	Reference	
Medium	1.53	0.98; 2.39	0.063
High	**2.13**	1.35; 3.36	0.001
	*p* for linear trend <0.001		

^1^ Logistic regression analysis adjusted for sex, age, allocation to intervention group, school, maternal education, physical activity, zBMI, and quality of life at baseline. ^2^ High HRQoL was defined as a KIDSCREEN-10 score > 53.1 units. Statistically significant OR are shown in bold.

**Table 3 nutrients-14-05304-t003:** Prospective analysis the association between baseline adherence to the Mediterranean diet and health-related quality of life at follow-up in Spanish boys and girls (*n* = 1371) ^1^.

	HRQoL (KIDSCREEN-10) ^2^		
	Beta Coefficient	95% CI	*p* Value
KidMed score ^3^ (unit)			
Model 1 ^4^	0.678	0.459; 0.896	<0.001
Model 2 ^5^	0.550	0.323; 0.778	<0.001
Model 3 ^6^	0.320	0.101; 0.540	<0.001

^1^ Linear regression analysis. ^2^ KIDSCREEN-10 ranged from 25.4 to 83.8 with higher score indicating higher HRQoL. ^3^ KidMed score ranges from −4 to +8 with higher score indicating a higher adherence to the MD. ^4^ Adjusted for sex and age. ^5^ Adjusted for variables of model 1 and allocation to intervention group, school, maternal education, physical activity, and zBMI. ^6^ Adjusted for variables of model 2 and baseline HRQoL.

## Data Availability

Data and materials are available upon request to the corresponding authors.

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
