# Peer review of "Prospective Association between Adherence to the Mediterranean Diet and Health-Related Quality of Life in Spanish Children"

_nutrients, 2022, doi:10.3390/nu14245304_

Round 1
Reviewer 1 Report
The paper is well-written and the followed methodology is clearly explained. I have reviewed the manuscript, in which authors evaluated the prospective association between adherence to the Mediterranean diet and Health-Related Quality of Life (HRQoL) in Spanish children. The work seems to fall in line with the journal’s scope, providing insights on the adherence to the Mediterranean Diet among young individuals and examines on relatively big sample size. All questions asked in the introduction appear to be sufficiently addressed throughout the text.
Introduction, I suggest to add also the evidence on benefits of the Mediterranean diet.
Discussion, Please add a paragraph briefly elaborating the information included in the introduction and discussing possible/hypothesized mechanism through which the Mediterranean diet exert favourable effect against diseases (i.e vegetable-based DM rich in insaturated fat, polyphenols etc can be a sustainable and ideal model for cardiovascular disease and certain cancer) .
Discussion, line 175 I suggest to add also the following study as representative for Italy: DOI: 10.1080/09637486.2020.1751089
Author Response
Reviewer: 1
Comments to the Author
The paper is well-written and the followed methodology is clearly explained. I have reviewed the manuscript, in which authors evaluated the prospective association between adherence to the Mediterranean diet and Health-Related Quality of Life (HRQoL) in Spanish children. The work seems to fall in line with the journal’s scope, providing insights on the adherence to the Mediterranean Diet among young individuals and examines on relatively big sample size. All questions asked in the introduction appear to be sufficiently addressed throughout the text.
Reply: Thank you.
Introduction,I suggest to add also the evidence on benefits of the Mediterranean diet.
Reply: We added the sentence “Unlike the Western diet, the Mediterranean diet (MD) diet is rich in phytochemicals, unsaturated fat, fibers and antioxidant known to improve cardiometabolic health and reduce chronic inflammation, but a recent body of evidence has also shown its benefits for cognitive health and against the risk of Alzheimer's disease, dementia and depression [4–7]” and the following related evidence:
Article 1
Becerra-Tomás N, Blanco Mejía S, Viguiliouk E, Khan T, Kendall CWC, Kahleova H, Rahelić D, Sievenpiper JL, Salas-Salvadó J. Mediterranean diet, cardiovascular disease and mortality in diabetes: A systematic review and meta-analysis of prospective cohort studies and randomized clinical trials. Crit Rev Food Sci Nutr. 2020;60(7):1207-1227. doi: 10.1080/10408398.2019.1565281. Epub 2019 Jan 24. PMID: 30676058.
Article 2
Valeria Tosti, MD, Beatrice Bertozzi, PhD, Luigi Fontana, MD, PhD, Health Benefits of the Mediterranean Diet: Metabolic and Molecular Mechanisms, The Journals of Gerontology: Series A, Volume 73, Issue 3, March 2018, Pages 318–326, https://doi.org/10.1093/gerona/glx227
Article 3
Shafiei F, Salari-Moghaddam A, Larijani B, Esmaillzadeh A. Adherence to the Mediterranean diet and risk of depression: a systematic review and updated meta-analysis of observational studies. Nutr Rev. 2019 Apr 1;77(4):230-239. doi: 10.1093/nutrit/nuy070. Erratum in: Nutr Rev. 2019 Jun 1;77(6):454. PMID: 30726966.
Article 4
Aridi, Y.S.; Walker, J.L.; Wright, O.R.L. The Association between the Mediterranean Dietary Pattern and Cognitive Health: A Systematic Review. Nutrients 2017, 9, 674. https://doi.org/10.3390/nu9070674
- Discussion, Please add a paragraph briefly elaborating the information included in the introduction and discussing possible/hypothesized mechanism through which the Mediterranean diet exert favourable effect against diseases (i.e vegetable-based DM rich in insaturated fat, polyphenols etc can be a sustainable and ideal model for cardiovascular disease and certain cancer) .
Discussion, line 175 I suggest to add also the following study as representative for Italy: DOI: 10.1080/09637486.2020.1751089
Reply: Discussion has been improved to reflect more its favourable effects against diseases (added sentences in italic).
“Because the MD is rich in polyunsaturated and monounsaturated fat, phytosterols and water-soluble fibres it reduces cholesterol level and protect against cardiovascular diseases [5]. The MD diet also decreases oxidative stress, inflammatory markers, DNA damage and cell proliferation making it a powerful ally against cancer [30]. Regarding mental health, a few potential molecular mechanisms have been proposed. The first and second mechanisms that involve inflammatory state and redox balance are closely related and have been both described in patients with depression [31–33]. Pro-inflammatory diets are characterized by high intakes of refined carbohydrates and saturated fats and low intakes of antioxidants and poly and monounsaturated fats which may alter the redox balance [34]. This alteration can lead to a state of chronic cellular inflammation that affects neurotransmitters, neuroendocrine function and brain activity and triggers mental health disorders [35,36]. The MD, characterized by reduced saturated fatty acid and amino acid intakes and increased phytochemical and fiber intakes, has been shown to reduce oxidative stress and inflammatory markers which can have a protective effect on the brain [5,37,38]. The third mechanism implicates the gut microbiota. Animal models fed on a high fat diet or high calorie diet showed a shift in gut microbiota population associated with anxiety, poorer cognitive flexibility, and impaired social and object recognition indicating a potential relationship between diet, gut microbiota, and brain [39,40]. The MD increases gut microbiome diversity, gut homeostasis, and microbiota-derived metabolites. In the elderly, microbiota enrichment following a MD diet was positively associated with improved cognitive function and reduced inflammatory markers [5,41]. A fourth mechanism is a dysregulation of the hypothalamic-pituitary-adrenal axis (HPA) which increases the level of cortisol, a frequent marker of patients with depression [42]. Intervention studies with vitamin C supplementation, omega 3 fatty acids or polyphenol rich food have been found to reduce cortisol level [43–46]. Furthermore, in a population of primates, a MD diet reduced the cortisol response to stress compared to primates fed on a Western diet [47]. A fifth potential mechanism comes from the brain-derived neurotrophic factor (BDNF) which also seems to be lowered in depressive patients [48]. In the PREDIMED study, depressive patients randomized to a MD supplemented with nuts reported lower risks of low plasma BDNF concentrations compared to the control group [49]. A final mechanism involving the tryptophan-kynurenine metabolism has been described due to its involvement in the production of serotonin, but its regulation as well as the availability of tryptophan are not yet fully understood [50].
In the from the reviewer indicated position (line 175) of the paper we discussed the findings of other studies regarding the association between health-related quality of life and adherence to the Mediterranean diet. However, the recommended article deals with another topic, namely the relationship between weight status and adherence to the Mediterranean diet. Therefore, we respectfully decline the addition of this proposed reference.
Reviewer 2 Report
Q1: Abstract:
1. The innovation of the article research has not been well demonstrated, as well as the potential molecular mechanisms.
Q2: Introduction:
1. The introduction to the Mediterranean diet and the so-called unhealthy diet styles including merits and demerits were suggested to be supplemented, as well as the related references in recent three years.
2. The significance and necessity of the study were not reflected in the last part of this section.
Q3: Materials and Methods:
1. As shown in 2.1., this study was designed as a prospective study in 2012? And the results were obtained in 2014? I am not sure that whether the timeliness of the data affects the publication of the paper.
2. The scoring and standards of each content were suggested to be displayed in a table.
3. The KIDSCREEN-10 Item statements were closely related with Health-related quality of life, but I could not find any correlations between these statements and Mediterranean diet, such as (7) Have your parent(s) treated you fairly.
Q4: Results:
1. Table 1, Total N=137?
2. How can Maternal education affect the Health-related quality of life, and what is the relationship with Mediterranean diet?
3. How were the tertile 1, 2, 3 grouped, were they all get Mediterranean diet?
4. In Table 2, how could we determine whether the significant impacts the results were positive or negative? And the relevant remarkable results were suggested to be fully analyzed.
5. There were relatively few valid data for the article, which was recommended to be further supplemented.
Q5: Discussion:
1. This section needed to be further improved, line 166 to 256, all these statements were other researchers’ conclusions, and I could not find any original insights or analysis of the authors of this article, as well as the relationship with these results obtained by authors.
2. Many statements of this section can be moved to the introduction section.
3. It is recommended that this part could be fully modified to reflect the innovation and research significance of this article, as well as the contributions of this field.
Q6: Conclusions:
1. The presentation was too short, which failed to fully reflect the research results of this paper.
Q7: Institutional Review Board Statement:
1. The approval time and approval number needed to be supplemented.
Author Response
Q1: Abstract
- The innovation of the article research has not been well demonstrated, as well as the potential molecular mechanisms.
Reply: We have slightly modified the abstract to underline the reason of our analysis as follows: “Health-Related Quality of Life (HRQoL) is gaining attention in children and adolescents because it is an important outcome of their health status and well-being. Therefore, it is important to identify determinants for HRQoL. Currently, there is scarce and mainly cross-sectional evidence on the relationship between adherence to the Mediterranean diet and HRQoL in children and adolescents. Therefore, the objective of the present study was to assess the prospective association between adherence to the Mediterranean diet and HRQoL in Spanish children”.
However, we don’t believe that the abstract is the part of the manuscript to present possible, and not measured, underlying mechanisms of findings from an observational study.
Q2: Introduction:
- The introduction to the Mediterranean diet and the so-called unhealthy diet styles including merits and demerits were suggested to be supplemented, as well as the related references in recent three years.
Reply: Thank you, this has been done (added sentences in italic).
“In 1948, health was defined by the World Health Organisation as “a state of complete physical, mental and social well-being and not merely the absence of disease or infirmity” [1]. From this definition began to emerge the concept of health-related quality of life (HRQoL) which is used to assess the perception of an individual's physical and mental health [2]. Widely applied to assess the quality of life in patients with chronic illnesses, it is also now an instrument of interest in the general population. In children and adolescents, perceived general well-being, including physical and psychological well-being, social interaction, and school performance, is particularly important for enjoying a fulfilling life and growing into adulthood confidently [3]. Unlike the Western diet, the Mediterranean diet (MD) diet is rich in phytochemicals, unsaturated fat, fibers and antioxidant known to improve cardiometabolic health and reduce chronic inflammation, but a recent body of evidence has also shown its benefits for cognitive health and against the risk of Alzheimer's disease, dementia and depression [4–7]. Likewise, cross-sectional evidence indicated that greater adherence to a MD was correlated with higher HRQoL however, with regard to longitudinal studies, only one evaluated this relationship and found no significant association after controlling for confounding factors and baseline HRQoL [8]. While adolescence is a time of significant physiological and psychological changes, promoting a MD from childhood may improve their HRQoL and help them navigate this transition more easily. However, the current limited evidence is based mainly on cross-sectional evidence with its inherent limitations [9–15]. There is a need for more robust evidence which can be provided by prospective study designs [16]. Therefore, the objective of the present study was to assess the prospective association between adherence to the MD and HRQoL in Spanish children.”
- The significance and necessity of the study were not reflected in the last part of this section.
Reply: We modified sentences of the introduction to better reflect the necessity of the study (added sentences in italic).
“In 1948, health was defined by the World Health Organisation as “a state of complete physical, mental and social well-being and not merely the absence of disease or infirmity” [1]. From this definition began to emerge the concept of health-related quality of life (HRQoL) which is used to assess the perception of an individual's physical and mental health [2]. Widely applied to assess the quality of life in patients with chronic illnesses, it is also now an instrument of interest in the general population. In children and adolescents, perceived general well-being, including physical and psychological well-being, social interaction, and school performance, is particularly important for enjoying a fulfilling life and growing into adulthood confidently [3]. Unlike the Western diet, the Mediterranean diet (MD) diet is rich in phytochemicals, unsaturated fat, fibers and antioxidant known to improve cardiometabolic health and reduce chronic inflammation, but a recent body of evidence has also shown its benefits for cognitive health and against the risk of Alzheimer's disease, dementia and depression [4–7]. Likewise, cross-sectional evidence indicated that greater adherence to a MD was correlated with higher HRQoL however, with regard to longitudinal studies, only one evaluated this relationship and found no significant association after controlling for confounding factors and baseline HRQoL [8]. While adolescence is a time of significant physiological and psychological changes, promoting a MD from childhood may improve their HRQoL and help them navigate this transition more easily. However, the current limited evidence is based mainly on cross-sectional evidence with its inherent limitations [9–15]. There is a need for more robust evidence which can be provided by prospective study designs [16]. Therefore, the objective of the present study was to assess the prospective association between adherence to the MD and HRQoL in Spanish children.”
Q3: Materials and Methods:
- As shown in 2.1., this study was designed as a prospective study in 2012? And the results were obtained in 2014? I am not sure that whether the timeliness of the data affects the publication of the paper.
Reply: To address the concerns of the reviewer, the study was originally designed as an intervention study, but in this article the data were used for a prospective secondary data analysis. This is a common practice, see for example the PREDIMED or PREDIMED Plus studies among others. Furthermore, we don’t see any argument why the time of publication should have an impact on the reported associations. Additionally, the underlying biological mechanisms are unlikely to be affected by time.
- The scoring and standards of each content were suggested to be displayed in a table.
Reply: Thank you, this has been added to the tables.
- The KIDSCREEN-10 Item statements were closely related with Health-related quality of life, but I could not find any correlations between these statements and Mediterranean diet, such as (7) Have your parent(s) treated you fairly.
Reply: The KIDSCREEN-10 in our study is the dependent variable and attempts to measure perceived health and quality of life. It is one of the most widespread used instruments and includes questions of different dimensions of health, as the Reviewer could check. And the proposal of our study is to assess if adolescents with greater adherence to the Mediterranean diet (independent variable), as part of a healthy life, better perceive their health and quality of life. The results of the Rasch analysis provided a unidimensional global HRQoL index. Therefore, each statement making up the overall KIDSCREEN-10 score cannot be treated independently.
Q4: Results:
- Table 1, Total N=137?
Reply: Thank you, we have now corrected this error.
- How can Maternal education affect the Health-related quality of life, and what is the relationship with Mediterranean diet?
Reply: Maternal education was included in the statistical analysis as a confounder. This confounder was significantly associated with both the exposure (positively, data not shown) and the outcome (positively, shown in table 1). This variable therefore meets the requirement for causal mediation analysis, which goes beyond the scope of the present work.
- How were the tertile 1, 2, 3 grouped, were they all get Mediterranean diet?
Reply: Tertile distribution of health-related quality of life was used to provide a better overview of the descriptive statistics. The results in the present study are derived from a secondary observational analysis. An intervention was initially performed in the sample therefore we adjusted for the allocation to intervention or control group.
- In Table 2, how could we determine whether the significant impacts the results were positive or negative? And the relevant remarkable results were suggested to be fully analyzed.
Reply: In this table, the results are presented in the form of odds ratio. The odds ratio greater than 1 and less than 1 had a positive and negative impact respectively on a high health-related quality of life. We don’t understand what the reviewer means with “fully analyzed”. The model was adjusted for all available potential confounders.
- There were relatively few valid data for the article, which was recommended to be further supplemented.
Reply: We don’t understand what the reviewer means with “few valid data”. We found throughout the literature only one longitudinal study which assessed the relationship between Mediterranean diet and health-related quality of life. This calls for more prospective evidence such as the findings provided in the present article. Additionally, the exposure and the outcome, in the present study, were measured with validated instruments.
Q5: Discussion:
- This section needed to be further improved, line 166 to 256, all these statements were other researchers’ conclusions, and I could not find any original insights or analysis of the authors of this article, as well as the relationship with these results obtained by authors.
Reply: We have restructured the discussion and added some explanations for different findings compared to our results when appropriated. But we respectfully disagree with the reviewer’s statement “I could not find any original insights or analysis of the authors of this article”. There are several cross-sectional articles on the present topic but only one prospective study. It is well known that evidence derived from cross-sectional data is limited and prospective studies are needed to verify the hypothesis created by cross-sectional analysis, and this is what we did. Indeed, the statistical analysis for this type of research is a standard procedure, especially when it didn’t include repeated measurements of the exposure variable.
- Many statements of this section can be moved to the introduction section.
Reply: We discussed our results with those of other authors. For this, the discussion section is the correct part of the manuscript.
- It is recommended that this part could be fully modified to reflect the innovation and research significance of this article, as well as the contributions of this field.
Reply: We have modified the discussion section as follows (modifications are in italic):
“In this study, we found that the KidMed score was prospectively associated with HRQoL after adjusting for gender, age, allocation to intervention group, school, maternal education, physical activity, zBMI and HRQoL at baseline. Through the literature, eight studies assessed the relationship between the KidMed index and the KIDSCREEN score from which one study had a longitudinal design while the others were cross-sectional [9–16]. Esteban-Gonzalo and colleagues reported in a longitudinal study design, a significant and positive association between adherence to the MD and HRQoL in adolescent girls but the statistical significance disappeared after further adjustment for HRQoL at baseline [16]. The authors found also a non-significant positive relationship between adherence to the MD and HRQoL [16]. The absence of statistical significance might be explained by the fact that they had a considerably smaller sample size in their strata compared to the present study, which in turn reduced statistical power. Additionally, Esteban-Gonzalo and colleagues provided results on subjective well-being based on another validated questionnaire, but these results were hardly comparable to ours because this questionnaire measured different aspects. Mixed results were also reported from cross-sectional studies [9–15]; five studies reported a positive significant association, one a positive non-significant association and one a negative non-significant association. Of the studies that found a significant positive association, all were concentrated in the Mediterranean region, including Greece, Spain, and Portugal [9–11,13,14]. Compared to our study, they were conducted in slightly older children, aged 11 to 18, and had a smaller sample size, with the largest reaching 956 participants [9–11,13,14]. Despite significant associations, most studies had little or no adjustment for confounders, with the exception of Evaristo and colleagues who adjusted for age, sex, pubertal stage, socioeconomic status, sleep duration and BMI [10]. A non-significant positive association was found in a Chilean study but compared to our results, children had a lower baseline KidMed score possibly suggesting a change in the Chilean diet [12]. Finally, Mitri and colleagues found a non-significant negative association however, in the multiple regression analysis, the KIDSCREEN-27 score had a variance inflation factor > 4 indicating a multicollinearity issue in the analysis [15]. Another prospective study of Australian adolescents aged 11-18 found a significant positive association between the MD diet and the emotional subscale of the PedsQL, but the questionnaire used to assess diet quality had not been previously validated [23].
Furthermore, we found that baseline consumption of fruit, vegetables and pulses was also positively associated with HRQoL at follow-up while consumption of fast-food, candies, pastries and pasta or rice reported the inverse relationship. To our knowledge, no prospective study has assessed these eating behaviors with HRQoL, nevertheless a few cross-sectional and cohort studies using other HRQoL-related questionnaires have provided insight [24–28]. With regard to fruit and vegetables intakes, a British study found that adolescents that consumed the optimal intake of fruit and vegetables (5 portions per day) had better score in physical functioning, emotional functioning, social functioning, psychosocial health summary score and total score of the PedsQL after adjusting for covariates [24]. This association remained true during summertime for emotional functioning and social functioning although in both time periods the magnitude was small [24]. Another study found that fruit and vegetables servings per day were positively associated to the social functioning subscale of the PedsQL in a group of American adolescents, but the study had a relatively small sample size [25]. Regarding unhealthy foods, fast-food, candies, pastries were negatively associated with higher HRQoL. Likewise, a study reported a reduction in HRQoL utilities score in Australian adolescents that ate often compared to sometimes or rarely fast food or takeaway after adjustment [29]. We did not find any association between skipping breakfast and HRQoL, but two Japanese cohort studies indicated a negative association [26,27]. These studies used the COOP chart questionnaire and the authors stated that it had been previously validated in children, but no abstract was found in either the Japanese pediatric population or other pediatric populations [26,27]. With respect to dairy consumption, a 5-year cohort study found that Australian adolescent boys with a high consumption of yoghurt had higher PedsQL’s psychosocial health summary and school functioning scores at follow-up but this study presented a high attrition rate (>50%) [28].”
Q6: Conclusions:
- The presentation was too short, which failed to fully reflect the research results of this paper.
Reply: Thank you, the conclusion was modified to fully reflect the research results:
“In conclusion, adherence to the Mediterranean diet was predictive of HRQoL in Spanish children. The consumption of fruits, vegetables and legumes, characteristic of the Mediterranean diet, was positively associated with high HRQoL, while the consumption of fast food and pasta or rice, more characteristic of the Western diet, was negatively associated with high HRQoL. Further prospective studies are needed to confirm this result.”
Q7: Institutional Review Board Statement:
- The approval time and approval number needed to be supplemented.
Reply: We have added the requested information: “Ethics Committee (CEIC - Parc de Salut Mar, approval number: (2011/4296/I) Barcelona, Spain).
Round 2
Reviewer 2 Report
Thanks for these modifications and explanations, I have no further questions.